# The Role of Musculoskeletal Ultrasound in Biologic Drug Tapering and Relapse Monitoring: Findings from a One-Year Prospective Study in a Cohort of Rheumatoid Arthritis Patients in Sustained Clinical Remission

**DOI:** 10.3390/diagnostics15141753

**Published:** 2025-07-10

**Authors:** Zguro Batalov, Tanya Sapundzhieva, Konstantin Batalov, Rositsa Karalilova, Anastas Batalov

**Affiliations:** 1Medical Faculty, Medical University, 4000 Plovdiv, Bulgaria; tanya.sapundzhieva@mu-plovdiv.bg (T.S.); k.batalovv@gmail.com (K.B.); karalilova@gmail.com (R.K.); abatalov@hotmail.com (A.B.); 2Rheumatology Clinic, University Hospital ‘Kaspela’, 4000 Plovdiv, Bulgaria; 3Rheumatology Department, University Hospital ‘Pulmed’, 4000 Plovdiv, Bulgaria

**Keywords:** ultrasound, Doppler, remission, rheumatoid arthritis, biologic therapy, tapering

## Abstract

**Objectives:** To assess the role of musculoskeletal ultrasound (MSUS) in selecting patients with rheumatoid arthritis (RA) in sustained clinical remission, suitable for tapering of biologic therapy (BT), and monitoring for a subclinical relapse. **Methods:** In this prospective study, seventy-eight patients with RA in sustained Disease Activity for twenty-eight joints (DAS28) clinical remission underwent ultrasound (US) examination of twenty-two joints (bilaterally wrists and metacarpophalangeal and proximal interphalangeal joints). US assessment was performed on gray scale ultrasound (GSUS) and power Doppler US (PDUS) to select patients in imaging remission, defined as a total PD score of synovitis = 0. Group 1 consisted of patients in clinical and imaging remission, in which tapering of BT was done through spacing of the Tumour Necrosis Factor Alpha (TNF-α) blocker. Group 2 consisted of patients only in clinical remission (PDUS > 0), who continued standard therapy. Clinical and US assessment was done at months 6 and 12, and the rate of a clinical (defined as DAS28 ≥ 2.6) and an US relapse (PDUS score ≥ 1) was recorded. **Results:** Thirty-eight patients were in clinical and US remission (group 1) and forty patients only in clinical remission (group 2). At month 6, 26% of patients in group 1 and 10% in group 2 experienced a clinical and an US relapse, whereas 20% and 15% of them, respectively, only an US relapse. At month 12, 26% of patients in group 1 and 20% of patients in group 2 experienced a clinical and an US relapse, whereas 35% and 22% of them, respectively, only an US relapse. **Conclusions:** Real-world data show that MSUS is a useful tool to identify RA patients in sustained clinical remission appropriate for BT tapering. US monitoring could predict a clinical relapse and the need to re-escalate treatment in patients with subclinical US relapse during BT tapering.

## 1. Introduction

Musculoskeletal ultrasound (MSUS) is a valuable method in the management of patients with rheumatoid arthritis (RA) according to the European League Against Rheumatism (EULAR) recommendations—from predicting which patients with undifferentiated arthritis would progress to RA, establishing the diagnosis, assessing for structural damage, monitoring the effect of treatment, confirming the presence of remission to predicting disease outcome and guiding treatment decisions [1,2,3,4,5]. Remission represents a state of lack of symptoms and signs of the disease, halting of both the radiographic progression and the decline in functional status over time [2].

Nevertheless, real-world data shows that structural progression is experienced even by patients in clinical remission [3]. Therefore, the state of clinical remission, assessed by patient history, physical examination, and laboratory markers, does not reflect full absence of ongoing joint inflammation, which can only be visualized by imaging methods, such as magnetic-resonance imaging (MRI) and MSUS [5,6]. Multiple studies have demonstrated that MSUS is more sensitive than palpation in detecting synovial inflammation [7,8].

The persistence of active synovitis (power Doppler (PD) positive) in RA patients in clinical remission has been proven to have practical implications because it can predict the progression of the radiographic damage and the higher risk of a flare of the disease activity [9,10].

The question of treatment duration or possibilities for treatment tapering is a common occurrence in clinical practice, especially in patients that show low disease activity or are in clinical remission.

According to the 2023 European League Against Rheumatism (EULAR) recommendations for the treatment of RA, tapering of BT can be considered after achieving sustained remission and corticosteroids (CS) withdrawal, especially if treatment is combined with a conventional synthetic disease-modifying drug (csDMARD) [11]. The possibility of tapering or the discontinuation of the biologic therapy (BT) in patients in remission or low disease activity (LDA) should be considered after a careful estimation of the benefits (biologic agents’ side effects, economic benefits) against the harm (risk for a disease relapse).

The main risk when trying to taper BT is a relapse of the disease activity. This underlines the importance of knowing the factors associated with an increased risk of a relapse. Naredo et al. [12] found presence of synovitis on PDUS to be the strongest predictor for failure of BT tapering in RA patients in sustained clinical remission. Alivernini et al. also discovered that sonographic examination with PDUS selects patients who will remain in clinical remission after tapering and discontinuation of the biologic agent [13]. Iwamoto et al. found that total gray scale (GS) and PD scores for synovitis have a high predictive value for the risk of a relapse after BT tapering [14]. Marks et al. discovered that MSUS helps identify patients in remission who would benefit from TNF-blocker dose reduction and ensures safe monitoring of subclinical relapses of disease activity [15].

Different results were obtained in a study by Lamers-Karnebeek, published in 2017. US was found to be a predictor of flare in patients with a longstanding low disease activity only at the group level, but at the patient level, it had limited added value in addition to the clinical parameters [16].

To determine the presence of true remission, described as an absence of active synovitis, US may be useful due to its high sensitivity in detecting synovitis [5,6]. The definition of US remission greatly varies across different studies. Most authors prefer to define it in the complete absence of a PD signal from the assessed joints [17,18]. Others describe US remission as the state of absence of synovitis detected both by GS and PDUS, which is the most stringent definition of all and difficult to reach even in the era of available targeted and biologic DMARDs [19]. In between the abovementioned definitions for US remission is the one defined by Van der Ven et al. as not greater than grade one of synovitis detected by GS and the absence of a PD-positive synovitis [20]. The least stringent definition of an US remission was applied in a recently published study by Diamanti et al., who accepted imaging remission in the absence of a PD signal ≥ 2 in one target joint, or PD ≥ 1 in two target joints [21].

Variability in defining sonographic remission complicates clinical decision making. A standardized approach is necessary to translate sonographic findings consistently into clinical practice. Our study aimed to assess the role of US examination as a tool to identify RA patients in imaging remission, appropriate for BT tapering. Secondary goals were to find the rate of a sonographic and a clinical relapse at months 6 and 12 after BT tapering and to compare the rates of relapse in patients who tapered and those who continued standard drug therapy.

## 2. Patients and Methods

### 2.1. Patients

Between September 2023 and May 2024, seventy-eight RA patients were consecutively enrolled in this prospective study. The inclusion criteria were a diagnosis of RA according to the 2010 ACR/EULAR 2010 classification criteria for RA, a state of sustained remission, defined as maintaining DAS28 remission for at least 6 months [22,23]. We ensured all patients were involved in a shared decision-making process when considering BT tapering. All patients were treated with csDMARDs—methotrexate (MTX) 10–20 mg weekly or leflunomide 20 mg/day and biologic DMARDs (bDMARDs) as standard of care according to the treating rheumatologist. The following data were recorded for each patient: age, gender, disease duration, and current therapy.

Patients were divided into two groups as follows: group 1—patients found to be in both clinical and sonographic remission, and group 2—patients only in clinical remission. Patients in group 1 had their BT tapered. BT was tapered according to an agreed strategy implemented in clinical practice that is increasing the interval between doses by one-third for subcutaneous biologic agents—adalimumab every 3 weeks, etanercept every 10 days, certolizumab every 3 weeks. Patients from group 2 continued standard doses of the biologic drug. All patients underwent physical and US examination at baseline at months 6 and 12. The rate of a clinical (defined as DAS28 ≥ 2.6) and an US relapse (PDUS score ≥ 1) was recorded. All patients in group 1 who relapsed (clinically or sonographically) had their BT escalated to the standard dose before tapering.

### 2.2. Methods

*Physical examination.* The same joint assessor, blinded to the clinical, laboratory, and imaging data of the patients, examined the 28 joints (right and left proximal interphalangeal (PIP) and metacarpophalangeal (MCP) joints, wrist, elbow, shoulder, and knee joints) for the presence of joint tenderness and/or swelling. A visual-analogue scale (VAS) from 0 to 10 cm was used to record the intensity of the joint pain. Disease activity was documented at baseline, months 6 and 12, using the DAS28 index.

*Laboratory tests.* At the baseline visit, levels of IgM-Rheumatoid factor (IgM-RF, positive if >20 U/L) and anti-citrullinated protein antibodies (ACPA, positive if >20 U/L) were documented. The level of C-reactive protein (CRP) (increased if >6.0 mg/L) was documented at all three visits.


*Ultrasound protocol*


All US joint examinations were perfomed by the same rheumatologist, who was blinded to the clinical and laboratory data of the participants, minimizing intra-observer variability. The same US machine, Esaote, MyLabX7 (Genova, Italy), possessing a 10–18 MHz linear transducer, was used throughout the study period. Twenty-two joints (right and left wrist, 1–5 MCP, 1–5 PIP joints) were assessed using both GSUS and PDUS. The following settings were adjusted: for the GSUS with a frequency of 18 MHz, gain varied according to the type of joint; for the PDUS with a frequency of 9.1 MHz and a pulse repetition frequency (PRF) of 500 Hz, PDUS gain was on average 50%, with slight variability in the different joint regions; the wall filter was maintained low. The gain of the PDUS was fixed during the three US examinations of the patient.

The scoring of GS and PD synovitis in a semi-quantitative way from 0 to 2 was performed according to the published 2017 EULAR-OMERACT consensus, ensuring optimal inter-observer reliability with previous US studies [24,25]. A dorsal scan was used for the assessment of both GS and PD synovitis. Two scores were obtained for each patient during each visit—the sum of scores of GS synovitis from all of the joints, a total GSUS score, one for PD synovitis, and a total PDUS score, respectively. Emphasizing the predictive value of PDUS for relapse, sonographic remission was defined as a total PDUS score = 0 [12,26]. The duration of the US scanning procedure, including reporting the findings, was 20–25 min.


*Statistical analysis*


Data analysis was performed using IBM SPSS Statistics for Windows, Version 28.0 (IBM Corp., Released 2021, Armonk, NY, USA). A power analysis was conducted for a comparison of proportions between two independent groups with sample sizes of 38 and 40 (total n = 78). We assumed a moderate effect size (Cohen’s H = 0.30) at a significance level (α) of 0.05. The estimated statistical power was approximately 0.80. This suggested an approximate 80% likelihood of identifying a true difference between the groups, assuming such a difference is present in the population. The chance of a Type II error was around 20%. Continuous variables were assessed for normality using the Shapiro-Wilk test. When normal distribution was confirmed, data were summarized through the mean and standard deviation (SD), and between-group comparisons were conducted using the independent-samples *t*-test. In cases of non-normal distribution, the median and interquartile range (IQR) were reported, and the Mann–Whitney U test was applied for between-group comparisons. Categorical variables were expressed as counts and percentages (%). To examine associations between them, Fisher’s exact test was used for dichotomous variables. For variables with more than two categories, the chi-square test and z-tests with Bonferroni adjustments were applied for pairwise comparisons. When expected frequencies in some cells were less than 5, the Monte Carlo method was used alongside the chi-square test, with a 99% confidence interval. To assess diagnostic agreement between the clinical and sonographic methods, we calculated the Kappa coefficient and the area under the receiver operating characteristic curve (ROC AUC). Z-tests were performed to compare proportions. All statistical analyses were conducted using a 5% significance level for Type I error, with results considered statistically significant at *p* < 0.05.

## 3. Results

### 3.1. Background Information About the Patients

The study included seventy-eight RA patients in DAS28 remission, aged 39 to 73, with a median age of 59.15 years. All patients were followed-up until month 12, with no missing data due to patients lost to follow-up. Of these, 79.50% (n = 62) were women and 20.50% (n = 16) were men. The median disease duration was 12.50 years, and the median remission duration was 26 months. At inclusion, 80.80% were ACPA-positive, and 65.40% were IgM-RF-positive.

At baseline, patients were divided into two groups: Group 1 comprised 38 patients (48.7%) who were in both clinical and sonographic remission, whereas Group 2 included 40 patients (51.3%) who were in clinical remission only. The groups did not differ significantly in terms of age (*p* = 0.723), gender distribution (*p* = 1.000), disease duration (*p* = 0.920), remission duration (*p* = 0.795), ACPA positive rate (*p* = 1.000), IgM-RF positive rate (*p* = 0.639), ACPA and IgM-RF negative rate (*p* = 0.811), and DAS28 median scores (*p* = 0.238) (Table 1).

### 3.2. Clinical and Sonographic Findings at the 6th Month

The remission and relapse rates between the two groups did not reveal significant differences at the 6th month of follow-up, as indicated in Table 2.

At month six, of the 38 patients in Group 1, 52.6% (n = 20) maintained the baseline status of both clinical and sonographic remission, while 26.3% (n = 10) experienced both a clinical and a sonographic relapse. The agreement between clinical and sonographic assessments was statistically significant, though relatively modest, as indicated by a Kappa coefficient of 0.568 and an AUC of 0.778. A discrepancy between the two diagnostic methods was observed in 21.1% (n = 8) of the patients, who experienced a sonographic relapse while being in clinical remission. This rate of false-negative diagnosis by clinical examination was statistically significant, as shown by the z-test for proportions (z = 2.99, *p* = 0.002).

In Group 2, at the six-month follow-up, 75.0% (n = 30) of the patients were in both clinical and sonographic remission, 10.0% (n = 4) experienced both a clinical and a sonographic relapse, and 15.0% (n = 6) only a sonographic relapse. Discordant findings between sonographic and clinical results accounted for 15.0% false negatives. This discordance was statistically significant (z = 2.54, *p* = 0.01). The overall level of agreement between clinical and sonographic assessments was statistically significant, though relatively modest, as indicated by a Kappa coefficient of 0.500 and an AUC of 0.700.

The level of agreement between sonographic and clinical findings was comparable for both patient groups, indicating that the clinical method missed some cases, with false negatives ranging from 15% to 21%. The comparison of the area under the curve revealed no statistically significant difference between the two groups (*p* = 0.937).

### 3.3. Clinical and Sonographic Findings at the 12th Month

At the twelfth month follow-up, the rates of relapse and remission in both groups did not exhibit significant differences (Table 3). Of the 20 patients in Group 1 who were found to be in both clinical and sonographic remission at the sixth month, 15 maintained this status, while 5 experienced a sonographic relapse only. Thus, the proportion of patients with both clinical and sonographic remission decreased to 39.40%.

Meanwhile, the number of patients with a sonographic relapse only increased from 8 at month six to 13 at month 12, resulting in a 34.20% false negative rate when compared to clinical assessment. The discordance between the two methods was statistically significant (z = 3.96, *p* < 0.001). The weighted Kappa value of 0.377 and the AUC of 0.717 indicate an acceptable, yet somewhat modest, concordance between clinical and sonographic evaluations.

In Group 2, at the 12th month, among the 30 patients who were in clinical and sonographic remission at the sixth-month follow-up, 23 (57.50%) retained the same status. Eight of them (20.00%) experienced both a clinical and a sonographic relapse. Three patients had a sonographic relapse only, raising the total number of sonographic relapses from six to nine (22.50%) by month twelve. The clinical assessment showed a false positive rate of 22.50%, which was statistically significant (z = 3.18, *p* = 0.001). The level of agreement between the clinical and sonographic methods was comparable to that observed at the 6th month, as indicated by a weighted Kappa of 0.505 and an AUC of 0.735 (Table 3).

Overall, the level of agreement between sonographic and clinical findings was similar in the two groups, showing a false negative rate between 34.20% and 22.50%. The comparison of the AUC values showed a lack of significant difference (*p* = 0.981).

Figure 1 and Figure 2 summarize the relapse rates based on clinical and sonographic assessments for each group over time.

In Group 1, the concordant clinical and sonographic relapse rate remained stable at 26.30% (n = 10) between the 6th and 12th months. However, the sonographic relapse rate increased from 21.10% (n = 8) to 34.20% (n = 13) during the same period.

In Group 2, simultaneous clinical and sonographic relapse was observed in 10% (n = 4) of the patients at month six and in 20% (n = 8) at month twelve. The sonographic relapse rate increased from 15.00% (n = 6) to 22.50% (n = 9).

In patients who experienced BT tapering failure—defined as either a clinical or a sonographic relapse—the standard dosage of BT was reinstated by their usual consulting rheumatologist. Among the 23 patients in Group 1 who relapsed following BT tapering, 12 patients (52%) returned to DAS28-defined remission, 8 patients (35%) achieved low disease activity, and 3 patients (13%) had moderate disease activity after treatment re-escalation. None of the patients required a change of the biologic agent following BT tapering.

## 4. Discussion

In the last years, the high financial costs of BT on health care systems have led to a great interest in optimizing BT by either withdrawing or tapering these drugs in RA patients in sustained clinical remission [27,28]. Evidence exists that MSUS detects persistence of synovitis on GS in almost all RA patients in clinical remission, of which many exhibit activity of the synovitis evidenced by the presence of a positive PD signal [9,29,30]. The clinical implication is in the effect of PD-positive synovitis over structural progression and risk of a disease flare [9,10].

There are several studies that explore the effects of BT tapering and withdrawal on disease control. Among these studies, there is considerable heterogeneity in subjects’ baseline disease duration and activity, as well as the protocols used for modification of bDMARD dosing. Tapering regimens also vary with different medication formulations and the posibility or lack there of to modify doses or dose frequency [31].

The primary objective of this prospective study was to assess the value of MSUS as a tool in selecting the appropriate patients in sustained clinical remission in whom BT tapering will be successful. This can be achieved due to the potential of US to find RA patients with the deepest remission (both clinical and imaging), who will have the lowest risk of relapse after BT tapering, thus being a step toward precise medicine in rheumatology.

For this study we used a uniform approach towards BT tapering regimens, applicable for all used subcutaneous bDMARDs. We used a PD-ultrasound score of 0 (PDUS = 0) as our benchmark definition of sonographic remission, guided by the predictive value of PD activity for disease relapse and the need for a strict cut-off for detection of subclinical synoviitis.

Evidence shows that approximately 20% of RA patients maintain DMARD-free remission [22]. In our study population at 6 months, 20% of patients from group 1 presented BT tapering failure because of a sonographic relapse, 25% because of a clinical and a sonographic relapse. At month 12, 35% of patients in group 1 presented a sonographic relapse and 25% of patients a clinical and a sonographic relapse. Overall, our results are relatively similar to those from other studies regarding BT tapering [13,14,15]. In addition, our study showed that BT tapering was non-inferior to usual care—the proportion of patients with a relapse at months 6 and 12 was similar in patients who tapered BT and who continued standard therapy. This finding was in agreement with the study of van Herwaarden et al., who found that the dose reduction strategy of adalimumab or etanercept to treat RA is non-inferior to usual care about major flaring, while resulting in the successful dose reduction or stopping in two-thirds of patients [32]. Our results suggest that the combination of clinical and US assessment may help identify individuals most appropriate for BT tapering.

PDUS has been proven to be a predictor for failure of BT tapering in RA patients in sustained clinical remission [12,13,14,15]. A systematic review from 2022 found that MSUS has a higher predictive ability for a relapse of the disease activity as compared to physical examination of RA patients in clinical remission [33].

Our results suggest that before considering BT tapering in RA patients in sustained clinical remission, US assessment (and in particular PDUS) may be useful in confirming the absence of Doppler-positive synovitis to indicate that the patient is appropriate for this therapeutic strategy. We decided to make treatment decisions considering PD-positive synovitis only, ignoring GS changes because their significance in later RA remains uncertain [34]. We chose to include only hand joints as US of the hand has been proven to be sufficient to detect subclinical inflammation in RA patients in remission [35]. The addition of PDUS identified a total of 13 patients in group 1 and 9 patients in group 2 in DAS28 remission, with subclinically active disease. Similar results were obtained by Marks et al. [15]. This finding is important as it reveals a subclinical relapse before the clinical exacerbation of the disease occurs. For real-world patient care, that puts an emphasis on the importance of US examination when following-up patients in clinical remission, enabling dose escalation early and preventing loss of remission.

Another important finding from our study, which has practical implications, is that following treatment re-escalation after a relapse of disease activity in patients who had tapered BT, 87% of patients restored the state of remission or low disease activity. Change of the biologic agent was not needed in any of the patients. These results are in agreement with the study of Marks et al. [15].

The long-term prospects for maintaining remission after BT tapering remain uncertain. While BT tapering might be possible within time-limited clinical studies, the feasibility of translating such clinical trial results to daily rheumatology practice over the longer term remains to be found.

Our study has several limitations. First, the population sample with RA was relatively small, and only patients on subcutaneous TNF-blocker were recruited. Second, only one tapering method was used—increased interval between doses. Third, inter- and intra-observer agreement for the sonographic examination was not tested. Furthermore patient-reported outcomes (PROs), including pain, physical function, and quality of life, were not included in this analysis but could contribute to future studies to provide a comprehensive evaluation of treatment impacts. Another limitation is the lack of monitoring of the radiographic progression in both treatment groups to assess whether there is a difference in the rate of structural damage in both treatment strategies—standard therapy and tapered BT. The last limitation is the lack of a comparison of the US-detected synovitis with a reference imaging modality, such as MRI.

## 5. Conclusions

MSUS may aid rheumatologists in identifying RA patients in sustained clinical remission who are suitable for BT tapering, as well as in monitoring for disease relapse. Larger-scale research is needed to further define the added value of ultrasound in BT tapering.

## Figures and Tables

**Figure 1 diagnostics-15-01753-f001:**
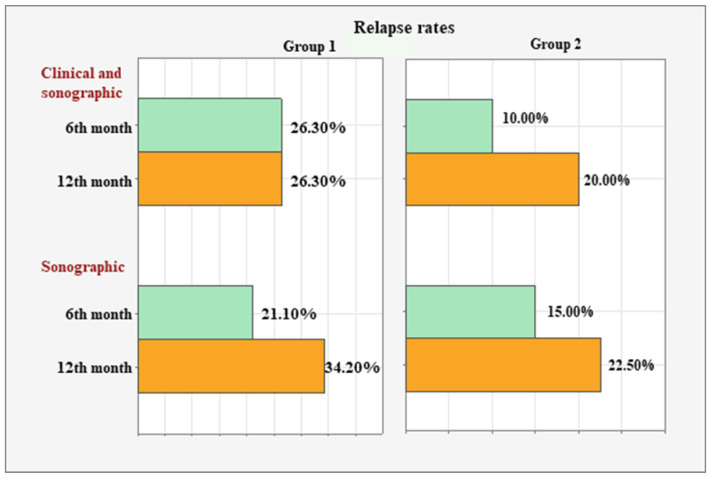
Clinical and sonographic relapse rates in both groups at months 6 and 12.

**Figure 2 diagnostics-15-01753-f002:**
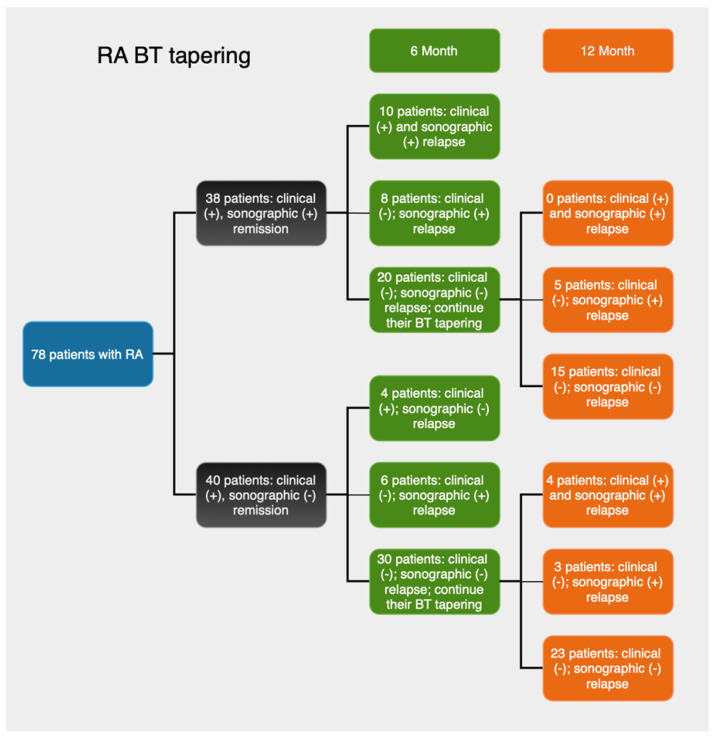
Summary of study design and clinical and sonographic findings at the sixth- and twelfth-month follow-ups.

**Table 1 diagnostics-15-01753-t001:** Patients’ demographic and clinical data at baseline.

Variables	Total (n = 78)	GROUP		*p*-Value
		**Group 1 (n = 38)**	**Group 2 (n = 40)**	
Age				
Median (IQR)	59.15 (13.35)	58.80 (14.52)	59.85 (13.05)	0.723 ^*U*^
Min–Max	39–73	43–73	39–72	
Gender n (%)				
Female	62 (79.50%)	30 (78.90%)	32 (80.00%)	1.000 ^*f*^
Male	16 (20.50%)	8 (21.10%)	8 (20.00%)	
Disease duration				
Median (IQR)	12.50 (10.00)	11.50 (12.25)	13 (8.00)	0.920 ^*U*^
Min–Max	3–26	3–26	3–23	
ACPA positive n (%)	63 (80.80%)	31 (81.60%)	32 (80.00%)	1.000 ^*f*^
IgM-RF positive n (%)	51 (65.40%)	26 (68.40%)	25 (62.50%)	0.639 ^*f*^
ACPA and IgM-RF negative n (%)	3 (3.80%)	1 (2.60%)	2 (5.00%)	0.811 ^*f*^
DAS28				
Median (IQR)	2.30 (0.40)	2.30 (0.30)	2.20 (0.40)	0.238 ^*U*^
Min–Max	1.50–2.50	1.50–2.50	1.60–2.50	
Remission duration (months)				
Median (IQR)	26 (24.25)	26.50 (23.50)	24.50 (27.00)	0.795 ^*U*^
Min–Max	7–50	7–48	7–50	

IQR—Interquartile Range; ACPA—Anti-citrullinated protein antibodies; IgM-RF—Immunoglobulin M-Rheumatoid Factor; DAS28—Disease activity score in 28 joints. ^*U*^—Mann-Whitney U; ^*f*^—Fisher’s exact test.

**Table 2 diagnostics-15-01753-t002:** Rates of remission and relapse at the sixth month.

Diagnosis	Group 1	Group 2	*p*-Value
	**(n = 38)**	**(n = 40)**	
Clinical and sonographic remission	20 (52.60%)	30 (75.00%)	0.090
Clinical and sonographic relapse	10 (26.30%)	4 (10.00%)	0.865
Sonographic relapse	8 (21.10%)	6 (15.00%)	0.949
Weighted Kappa	0.568	0.500	
95% CI	0.328 to 0.807	0.180 to 0.819	
AUC	0.778	0.700	
SE	0.06	0.08	0.937
95% CI	0.614 to 0.896	0.535 to 0.834	

CI—Confidence Interval; AUC—Area Under the Curve.

**Table 3 diagnostics-15-01753-t003:** Rates of remission and relapse at the twelfth month.

Diagnosis	Group 1 (n = 38)	Group 2 (n = 40)	*p*-Value
Clinical and sonographic remission	15 (39.40%)	23 (57.50%)	0.909
Clinical and sonographic relapse	10 (26.30%)	8 (20.00%)	0.954
Sonographic relapse	13 (34.20%)	9 (22.50%)	0.922
Weighted Kappa	0.377	0.505	
95% CI	0.163 to 0.592	0.258 to 1.000	
AUC	0.717	0.735	0.981
SE	0.05	0.06	
95% CI	0.548 to 0.851	0.572 to 0.862	

CI—Confidence Interval; AUC—Area Under the Curve, SE—Standard Error.

## Data Availability

The data presented in this study are available on request from the corresponding author due to privacy reasons.

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
