# Peer review of "The Role of Musculoskeletal Ultrasound in Biologic Drug Tapering and Relapse Monitoring: Findings from a One-Year Prospective Study in a Cohort of Rheumatoid Arthritis Patients in Sustained Clinical Remission"

_diagnostics, 2025, doi:10.3390/diagnostics15141753_

Round 1
Reviewer 1 Report
Comments and Suggestions for Authors
The Role of Musculoskeletal Ultrasound in Biologic Drug Tapering and Relapse Monitoring: Findings from a One-Year Prospective Study in a Cohort of Rheumatoid Arthritis Patients in Sustained Clinical Remission
The manuscript covers an interesting topic. However, some points need to be addressed
Abstract
- The abstract is clear, but it might be more detailed about how many patients were included and what the exact criteria were for sonographic and clinical remission. For instance, stating that DAS28 should be used for clinical remission and that there should be no power Doppler signal for sonographic remission would make the methods clearer.
- The abstract should better explain how this study builds on what has already been written on the topic. For instance, emphasizing that it offers prospective, real-world data on the application of MSUS in BT tapering and relapse monitoring would accentuate its significance.
Introduction
- The introduction may briefly talk about how the study's results could be used in clinical practice, including how MSUS could be used in routine treatment for RA patients who are thinking about tapering off BT. This would assist readers understand how important research is in the actual world.
- The introduction has a lot of information and references, but it could turn into a mini review. Think about cutting down on some parts so that the most important literature is the only thing that gets your attention, and the story stays short and to the point, leading directly to the study's goals.
- The introduction with how different people defines US remission, which is a crucial point. But it might be made explicit how this diversity affects clinical decision-making and why a uniform strategy is necessary.
Materials and methods
- The text indicates that a single rheumatologist conducted all ultrasound examinations, so ensuring uniformity; nonetheless, it does not account for inter-observer variability. Adding a remark about inter- and intra-observer reliability for ultrasound scoring would make the results more reliable, especially since interpreting ultrasound is subjective.
- The study has 78 patients, which is appropriate for a single-center, prospective investigation. A quick explanation of the sample size (like a power calculation) would, however, make the methods clearer and assist readers figure out how likely it is that type II errors will happen.
- The definition of sonographic remission (total PDUS score = 0) is unambiguous and consistent with contemporary research. The manuscript should quickly explain why this definition was chosen above others, especially if the introduction lists several different definitions.
Results
- The text presents Kappa coefficients and AUC values; nevertheless, it would be beneficial to include a concise interpretation of these metrics (e.g., “moderate agreement” for Kappa, “good diagnostic accuracy” for AUC) to assist readers who may not be acquainted with these statistics.
- The publication does not clearly say if there was any missing data or if there were any losses to follow up during the study period. Adding this information would make the results clearer and stronger.
- The study emphasizes clinical and sonographic outcomes, although it omits patient-reported outcomes such as pain, function, and quality of life. If this data are accessible, their inclusion would yield a more thorough evaluation of the effects of tapering and relapse on patients' lives.
Discussion
- The discussion provides a good backdrop for the study; however it might make the new contributions of the current work clearer. For instance, the publication might explain how this study builds earlier work in the field, especially as there is no clear definition of US remission and there is still a lot of discussion regarding how useful MSUS is in making clinical decisions.
- The discussion could further elucidate the clinical implications of the observed discordance between clinical and sonographic evaluations, particularly the elevated incidence of subclinical recurrence identified by MSUS. There may be a more in-depth discussion of what this means for patient care and the possibility of earlier intervention.
- The study juxtaposes its findings with those of Marks et al. and van Herwaarden et al. However it may benefit from a more systematic comparison with other pertinent research, especially those employing alternative tapering procedures or diverse demographics. This would help readers comprehend how the results can be used in other situations.
- The discussion suggests that additional research and longer follow-up are needed, but it should be clearer about what research should be done first. For instance, the necessity for multicenter research, uniform tapering methods, cost-effectiveness evaluations, and the incorporation of patient-reported outcomes might be emphasized.
Author Response
Thank you very much for taking the time to review this manuscript. Your critical remarks are of great
value and will help us improve the quality of our work. Please find the detailed responses below and
the corresponding corrections highlighted in the re-submitted files.

Reviewer 2 Report
Comments and Suggestions for Authors
Thank you for the opportunity to review your manuscript, “The Role of Musculoskeletal Ultrasound in Biologic Drug Tapering and Relapse Monitoring: Findings from a One-Year Prospective Study in a Cohort of Rheumatoid Arthritis Patients in Sustained Clinical Remission”
The aims was to assess the role of US examination as a tool to identify RA patients in imaging remission, appropriate for BT tapering. Secondary goals were to find the rate ofa sonographic and a clinical relapse at months 6 and 12 after BT tapering and to compare the rates of relapse in patients who tapered and those who continued standard drug therapy.
The manuscript presents a relevant topic with clinical applicability, featuring an appropriate design and patient follow-up. The authors have already identified the limitations regarding sample size, and the study does not present a sample size calculation. Additionally, I did not find any mention of whether there were losses to follow-up during the study.
The manuscript is well-written, and within its complexity, the structure is well-organized. The conclusions are appropriate to the results.
Aspects to improve:
Abstract: The abstract is not structured and only presents the study objective. It should be restructured with the sections suggested by the journal.
Figure 2: The quality and legibility of Figure 2 should be improved. Furthermore, it should be placed within the Results section.
Author Response
Thank you very much for taking the time to review this manuscript. Please find the detailed
responses below and the corresponding corrections highlighted in the re-submitted files.
Comments 1: The manuscript presents a relevant topic with clinical applicability, featuring an appropriate design and patient follow-up. The authors have already identified the limitations regarding sample size, and the study does not present a sample size calculation. Additionally, I did not find any mention of whether there were losses to follow-up during the study. |
Response 1: We are grateful for the reviewer’s kind words about our work. They raise a couple of important issues which we have tried to address. Sample size calculation is added to the revised manuscript. Data regarding patients that are lost to follow up is included as well. |
Comments 2: Abstract: The abstract is not structured and only presents the study objective. It should be restructured with the sections suggested by the journal. |
Response 2: Thank you for pointing that out. There was a technical error with formatting on our part, which resulted in only the first paragraph of the abstract to be uploaded. A revised abstract was subsequently uploaded, which we hope is in accordance with the journal suggested format. |
Comments 3: Figure 2: The quality and legibility of Figure 2 should be improved. Furthermore, it should be placed within the Results section.
Response 3: Thank you for your suggestion. Figure 2 is redrawn with better legibility and moved to Results section.
Round 2
Reviewer 1 Report
Comments and Suggestions for Authors
The manuscript improved